# Room-temperature stabilizing strongly competing ferrielectric and antiferroelectric phases in PbZrO₃ by strain-mediated phase separation

Ziyi Yu [1,2], Ningbo Fan [3], Zhengqian Fu [1] ✉, Biao He[1], Shiguang Yan[1], Henghui Cai[1], Xuefeng Chen [1], Linlin Zhang[1], Yuanyuan Zhang[4], Bin Xu [3], Genshui Wang [1] ✉ & Fangfang Xu [1,2] ✉

PbZrO₃ has been broadly considered as a prototypical antiferroelectric material for high-power energy storage. A recent theoretical study suggests that the ground state of PbZrO₃ is threefold-modulated ferrielectric, which challenges the generally accepted antiferroelectric configuration. However, such a novel ferrielectric phase was predicted only to be accessible at low temperatures. Here, we successfully achieve the room-temperature construction of the strongly competing ferrielectric and antiferroelectric state by strain-mediated phase separation in PbZrO₃/SrTiO₃ thin film. We demonstrate that the phase separation occurs spontaneously in quasi-periodic stripe-like patterns under a compressive misfit strain and can be tailored by varying the film thickness. The ferrielectric phase strikingly exhibits a threefold modulation period with a nearly *up-up-down* configuration, which could be stabilized and manipulated by the formation and evolution of interfacial defects under applied strain. The present results construct a fertile ground for further exploring the physical properties and applications based on the novel ferrielectric phase.

Antiferroelectric (AFE) oxides are attracting widespread attention for their promising applications in a wide range of areas including high-power capacitors, large-strain actuators, electrocaloric refrigeration devices, and negative capacitance electronic technologies[1–5]. Among this class of materials, PbZrO₃ (PZ) remains the most essential because it is considered as the prototypical AFE since Kittel's theoretical definition in 1951[6–8]. Based on extensive experimental and theoretical investigations, it is the consensus that the *Pbam* phase with antipolar ordering of *up-up-down-down* arrangement is the ground-state

configuration in PbZrO₃ and its isostructural counterparts, such as PbHfO₃[9–17]. To date, understanding the origin of AFE, establishing structure-property correlation, and developing new AFE applications are all based on such a simple antipolar model.

The common belief in antiparallel dipoles configuration, however, has been challenged recently by first-principles theoretical calculations, which suggest several different ferrielectric (FiE) configurations could be generated and appear more stable than the commonly accepted antiferroelectric arrangement, such as the 30-atom *Ima2* FiE

[1]State Key Laboratory of High Performance Ceramics and Superfine Microstructures & The Key Lab of Inorganic Functional Materials and Devices, Shanghai Institute of Ceramics, Chinese Academy of Sciences, Shanghai 200050, China. [2]School of Physical Science and Technology, ShanghaiTech University, Shanghai 201210, China. [3]Jiangsu Key Laboratory of Thin Films, School of Physical Science and Technology, Soochow University, Suzhou 215006, China. [4]Key Laboratory of Polar Materials and Devices, Ministry of Education, Department of Electronic Science, East China Normal University, Shanghai 200241, China. ✉e-mail: fmail600@mail.sic.ac.cn; genshuiwang@mail.sic.ac.cn; ffxu@mail.sic.ac.cn

phase and the 80-atom *Pnam* AFE-like phase[18–20]. Although these predicted phases exhibit very small energy difference from the *Pbam* AFE phase, the *Ima2* FiE phase is a special one because it has a surprisingly threefold modulation period while the most reported FiE phases have larger modulation period than the fourfold archetypal *Pbam* AFE phase[8,21]. In case there is no solid theoretical understanding towards ferrielectricity or antiferroelectricity so far and there exist many conflicting viewpoints, the experimental confirmation of these theoretically predicted phases in PZ will facilitate further understanding towards PZ. Recently, the observations in PZ single crystal and film undoubtedly demonstrate that the predicted threefold FiE phase can be achieved at room temperature, despite only focusing on translational boundaries and phase transition[22,23]. Thus, there is currently a strong desire to know how the threefold FiE coexists with AFE phase and whether the threefold FiE phase could be manipulated at ambient temperatures.

Phase separation (PS) is a well-known process bringing about formation of two-phase mixtures from a homogeneous state. As the famous spinodal decomposition for an example, it can occur even without a thermodynamic barrier. Thus, the energy barrier between the conventional antiferroelectric configuration and the theoretically proposed ferrielectric ground-state configuration may be overcame once the PS occurs. Starting from this point, here we demonstrate that the strain-mediated PS in PbZrO$_3$ film can be utilized for stabilizing the strongly competing states even at room temperatures. Such finding could provide a new opportunity to understand the diversity of configurational tailoring of dipoles ordering and provoke further progress in property innovation by utilizing the ferrielectric phase.

## Results and discussion

An optimum stress modulation is crucial for achieving phase transition via strain engineering in thin films. In general, this is realized by choosing an appropriate substrate with specific orientation and certain lattice misfit. It is noted that too tiny misfit strain can hardly induce PS while too large misfit strain may cause failure of epitaxial growth or even, film cracking. Table 1 lists the reported constructions of PZ films grown onto different substrates and their theoretical lattice misfit values. These constructions with misfit in a wide range of $-7.22\% \sim +0.72\%$ all succeeded in the fabrication of epitaxial thin films, but only the constructions of PbZrO$_3$/SrTiO$_3$ and PbZrO$_3$/ SrRuO$_3$/ SrTiO$_3$ thin film showed additional competing ferrielectric or ferroelectric phases embedded in AFE phase[23,24]. Therefore, in this work, we choose single crystal SrTiO$_3$ (STO) as the substrate and directly grow PZ thin films onto it by the chemical solution deposition method (CSD). Such setup results in $\varepsilon \approx -6.5\%$ lattice misfit and induces relatively large compressive strain. In addition, electron diffraction is also applied in the present study attempting to identify the possible PS in the PZ film thanks to much stronger scattering of electron beams than X-rays.

Firstly, we carried out first-principles calculations by using density functional theory (DFT) to investigate the stability of the conventional AFE phase and the theoretically predicted FiE phase under the strain within pseudo-cubic (100) (equivalent to (010)) and (001) plane. The AFE and FiE phases under (100) strain are designated as *a*-AFE and *a*-FiE, respectively, and those under (001) strain are designated as *c*-AFE and *c*-FiE, respectively. It can be seen that the AFE phase is generally more favored than the FiE phase under (100) strain (see Fig. 1a), while the FiE phase is unprecedentedly more favored under (001) compressive strain (see Fig. 1b). Despite both AFE and FiE have the polarization along $\pm[1\bar{1}0]$ direction on the (001) plane, the compressive stress may have different effects on the stability of "↓↓↑↑" configuration or "↓↓↑" configuration depending on the direction of stress with respect to the polarization direction. It seems that a compressive stress with its direction out of the polarization plane would favor the stability of the FiE phase (see Fig. 1b).

Then, PZ films were grown on the <100>-oriented STO substrate with three different thickness, i.e., 60, 160 and 180 nm, so as to further tailor the stress states through film thickness. The high-resolution X-ray diffraction (HRXRD) patterns (Fig. S1) and the reciprocal space mapping (RSM, Fig. S2) indicate all PZ films are well epitaxially grown. The comparatively weak intensity of XRD peaks for the PZ phase could be attributed to the existence of certain amount of pores in films as seen in the cross-sectional SEM images (Fig. S3). The evolution of P-E hysteresis loops and dielectric tunability of the three PZ films are presented in Fig. S4. It can be seen that the hysteresis loops demonstrate gradual occurrence of bulging shape near the zero-electric field upon increasing the film thickness. This bulging shape indicates increased remnant polarization in thicker film, which is most likely related to the presence of mixed-phases of AFE and FiE state. The selected-area electron diffraction (SAED) patterns in cross-sectional dimension (Fig. S5a–c) reveal that all PZ films exhibit a modulated structure along [110] direction as seen from the appearance of satellite reflections in-between main reflections. According to the extensive structural studies on PZ, these satellite reflections should be related with the modulation of Pb-cations displacement[13]. Of particular interest is that there are two different types of satellite reflections (marked by "T" and "Q" respectively) that simultaneously occur between main reflections (Fig. 2a–c, the main reflections were labeled by yellow arrows). Moreover, the thickness of PZ film can lead to a trade-off between the T-type and Q-type satellite reflections, which is evidenced by the evolution of intensity profiles in Fig. 2a–c and real-space dark-field (DF) images in Fig. S6.

**Table 1 | Comparison of theoretical strain misfit for PZ thin films on different substrates**

| Substrate | Lattice constant [Å] | | | Epitaxial relationship | Misfit ε [%][a] | Ref. |
|---|---|---|---|---|---|---|
| | $a_P$[b] | $b_P$ | $c_P$ | | | |
| LaNiO$_3$ | 3.87 | 3.87 | 3.87 | (001)$_P$ PZ \|\| (001) LNO | −7.22 | 44 |
| LaNiO$_3$ | 3.87 | 3.87 | 3.87 | (110)$_P$ PZ \|\| (110) LNO | −7.22 | 45 |
| SrRuO$_3$ | 3.928 | 3.928 | 3.928 | (100)$_P$ PZ \|\| (100) SRO | −5.74 | 24,46 |
| PbTiO$_3$ | 3.9 | 3.9 | 4.13 | [100] PZ \|\| [100]$_P$ PTO | −6.45 | 47,48 |
| Nb:SrTiO$_3$ | 3.9 | 3.9 | 3.9 | (111)$_P$ PZ \|\| (111) NSTO | −4.89 | 49 |
| BaZrO$_3$ | 4.19 | 4.19 | 4.19 | (100)$_P$ PZ \|\| (100) BZO | +0.72 | 27 |
| Pt | 3.923 | 3.923 | 3.923 | (100)$_P$ PZ \|\| (111) Pt | −5.86 | 50 |
| Pt | 3.923 | 3.923 | 3.923 | (111)$_P$ PZ \|\| (111) Pt | −4.39 | 26,51 |
| SrTiO$_3$ | 3.9 | 3.9 | 3.9 | [100] PZ \|\| [100]$_P$ STO | −6.45 | 23, this work |

[a]ε is the lattice misfit in a-axis of PZ caluclated from the equation: $\varepsilon = 2(a_{sub} - a_{PZ})/(a_{sub} + a_{PZ}) \times 100\%$; lattice constant of PbZrO$_3$: $a_P = b_P = 4.26$ Å, $c_P = 4.12$ Å[13].

[b]All Subscript P in this Table represent pseudocubic.

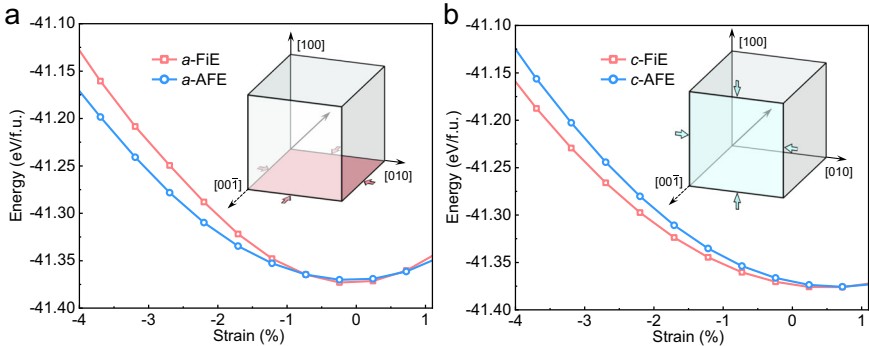

**Fig. 1 | DFT predicted stabilities of the AFE and FiE phases.** Total energies of the FiE and AFE phases as a function of epitaxial biaxial strains in (100) plane (**a**) and in (001) plane (**b**). The insets show the schematic diagram of applied biaxial strain, where both FiE and AFE phase have their dipole ordering in (001) plane along [110] or [1$\bar{1}$0] direction.

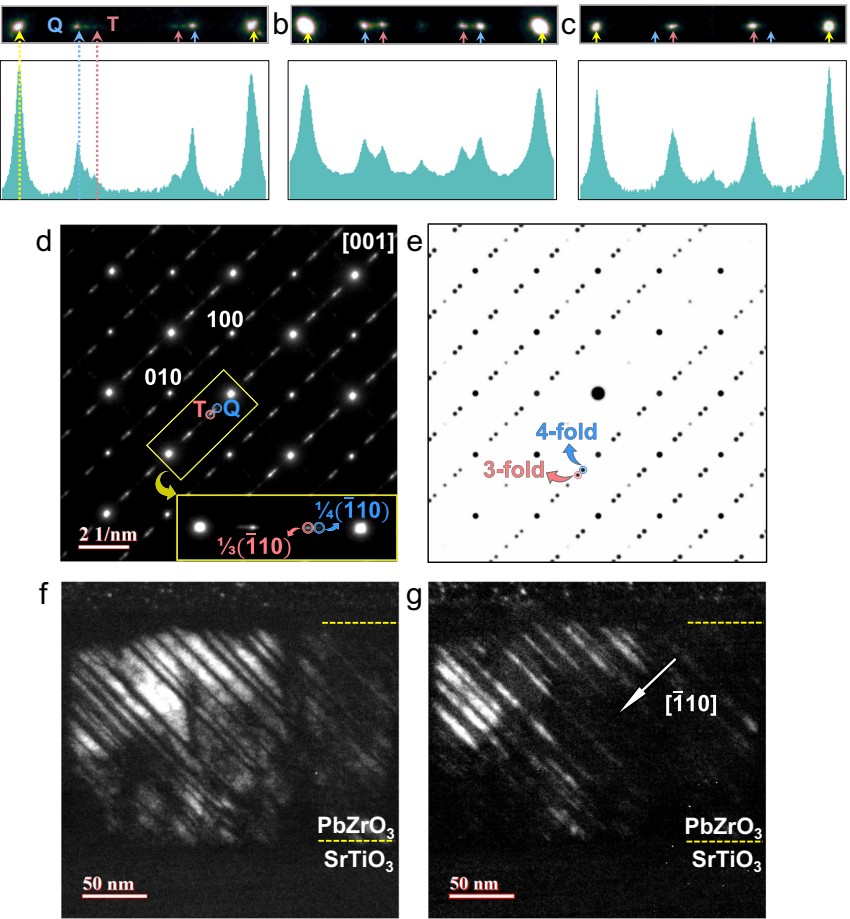

**Fig. 2 | Phase separation in PZ thin films. a–c** The magnified SAED patterns in Fig. S5 and the corresponding intensity profiles of T-type and Q-type satellite reflections for 60, 160 and 180 nm PZ films, respectively. **d, e** Experimental and simulated selected-area electron diffraction pattern of the mixed-phases, respectively. Secondary diffraction results in appearance of reflections at the crystallographically extinction sites on the experimental diffraction pattern. **f, g** Cross-sectional dark-field images of stripe-like mixed-phases in (100) PZ film on (100) STO substrate acquired using "T" and "Q" satellite reflection in **d**, respectively. **d–g** were obtained from 180 nm PZ film.

By measuring the distance between the satellite and main reflections (Fig. S7), it can be clearly seen that the T-type and Q-type satellite reflections give a triple and a quadruple superlattice with respect to the basic perovskite unit cell, respectively. This result implies the phase separation between AFE- and FiE-like phase is achieved because the well-known AFE phase and the theoretically predicted FiE phase in PZ precisely exhibit a quadruple and a triple modulated period, respectively. We then modeled two Pb-sublattices, i.e., the AFE phase of *up-up-down-down* configuration and the FiE phase of *up-up-down* configuration. The superimposed simulated SAED pattern (Fig. 2e) of these two Pb-sublattices is highly consistent with the experimental SAED pattern in Fig. 2d, confirming the occurrence of phase separation between AFE and FiE phase.

Figure 2f and g present the cross-sectional DF images using T-type and Q-type satellite reflections, respectively. The totally complementary bright-dark contrast between Fig. 2f and Fig. 2g indicates that the phase separation between AFE- and FiE-like phase is characterized by a nearly ordered array of parallel stripes with a width of 3–20 nm. Furthermore, the phase boundaries are habited on a (110) plane. Because the direction of Pb-displacements in PZ is along [110]$_P$, for the [100]-oriented films in the present study, the phase boundaries maintain 45° to minimize the elastic energy.

Antiferroelectric PZ ceramics usually show AFE multidomains along with high density of interior parallel antiphase domain boundaries (APBs) (see Fig. S8), while no PS has ever been observed. However, in the present PZ thin films, we observed few domains (see Fig. S9) but a widespread phase separation phenomenon in the form of nanosized intergrowth stripes in PZ/STO thin films. The ultrashort span of frequent phase separation hence high density of phase boundaries implies extremely large internal stress that has been generated during film growth.

Figure 3a shows the atomic-scale high-angle annular dark-field (HAADF) image of 180 nm PZ/STO thin film. Periodic misfit dislocations could be observed at the PZ/STO interface. By carrying out geometric phase analysis (GPA) with the $(0\bar{1}0)$ and (100) basic reflections, it can be seen that the compressive strain from the substrate is almost totally relaxed and there is no obvious stress gradient throughout the thin films (Fig. 3b, c). The misfit dislocations have relaxed majority of the compressive strain by noting the stress concentration at the dislocation cores (see Fig. 3b) along the PZ/STO interface. According to the RSM measurements (Fig. S2), the lattice parameters of AFE-like and FiE-like phase in 180 nm PZ are $a_p = b_p = 4.154$ Å, $c_p = 4.123$ Å and $a_p = b_p = 4.163$ Å, $c_p = 4.160$ Å, respectively. Thus, albeit slightly larger of the FiE unit cell, the difference of lattice parameters between the two phases is quite small in the present thin film, hence the interphase strain map shows only small contrast variation in Fig. 3b. When the strain maps were obtained by using the satellite reflections, the distribution of three and fourfold phases could be conveniently visualized as seen in Fig. 3e and f. The EDS elemental maps exhibit homogeneous elemental distribution in both AFE- and FiE-like phases, suggesting there is no chemical composition effect on the phase separation (Fig. S10). According to Fig. 3b and Fig. 3e, the GPA images demonstrate that both of FiE phase and AFE regions cover the misfit dislocations, implying that there is no direct correspondence between the nucleation sites of FiE (or AFE) phase and the cores of misfit dislocations. However, the local strain induced by these dislocations would influence the type and formation

of APBs[25], which indirectly intervene the stabilization and coexistence of FiE-like and AFE-like phases as will be discussed later.

Figure 4a shows the atomic-scale HAADF image containing a phase boundary between AFE- and FiE-like phases. The locations of the atomic columns were fitted by least squares estimation algorithm using MATLAB code[26] and then the Pb displacement was calculated by referring to the averaged surrounding Zr positions. The fitting of the atomic columns for Pb atoms in this work has a 95% confidence interval of 4.15 pm. It can be seen in Fig. 4b that the displacement of Pb cations demonstrates 3-fold and 4-fold modulated configuration in T-phase and Q-phase, respectively. The fast Fourier transformation (FFT) patterns (see the insets in Fig. 4a) further confirm that they are FiE- and AFE-like phase, respectively. The FiE-like structure highly conforms with the previous first-principles calculation by Aramberri et al.[18]. In addition, we found that the "↑↑↓" threefold ordering configuration could also show a rotation of dipoles by 45° in some local area (see Fig. S11). For the AFE phase, the strains arisen from the neighboring FiE phase and the substrate results in comparatively large deviation from the perfect "↑↑↓↓" ordering for Pb displacement. Figure 4c illustrates the profile of the inter-atomic distance along a (010) atomic plane and across the phase boundary. It can be seen that AFE- and FiE-like phases show different rules for the magnitude oscillation: non-symmetrical displacement of Pb in the FiE phase, i.e., one large displacement followed by two small displacement while fairly uniform and symmetrical displacement in the 4-fold AFE phase. This also indicates structural deviation of stress-induced room-temperature FiE-like phase structure from the theoretically predicted one in case of stress-free and at low temperatures. In addition, the peak value of the cell parameters in the FiE-like phase is larger than the one in the AFE-like phase.

To understand the specific nanoscale lamellar dual-phase intergrowth morphology, a schematic diagram is presented in Fig. 5. During the formation of modulated structures, many interfacial defects, i.e. APBs will spontaneously occur. The APBs have been found to expand on the (110) plane[25], while the lamellar crystallites of the threefold phases lay on the same crystallographic plane. These APBs can exhibit the "↑↑↓"-like ferrielectric configuration[25,27]. Meanwhile, according to the observed interfacial defect driven phase transition in PbZrO$_3$ single crystal materials[22], it is reasonable to speculate that these APBs could be the possible nucleation sites for the FiE phase. As shown in Fig. 5a, the APBs exhibit one unit of "↑↑↓" FiE-like configuration. During the preparation of thin film (especially at the cooling stage under compressive strain), the APBs are coarsened and the embryonic form of the FiE phase is generated (Fig. 5b). Eventually, the FiE-like phase further

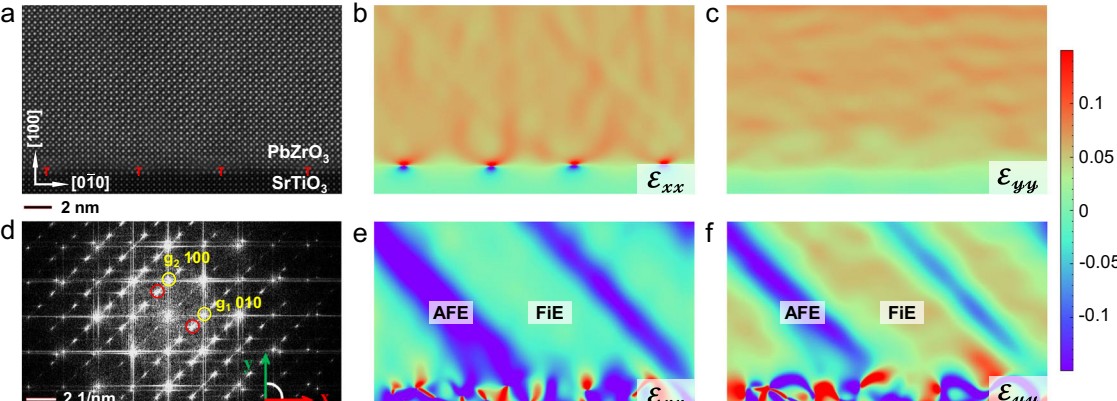

**Fig. 3 | Strain fields in the GPA analysis of 180 nm PbZrO$_3$ thin film. a** HAADF-STEM image of the cross-sectional PZ/STO thin film. (**b**, **c**) The corresponding maps of local strain fields compared with the STO substrate. **e, f** The corresponding maps of distribution of FiE and AFE phase. **d** The Fourier transform of image (**a**). The yellow circles mark the Gaussian masks for **b**, **c** and the red ones for **e, f**.

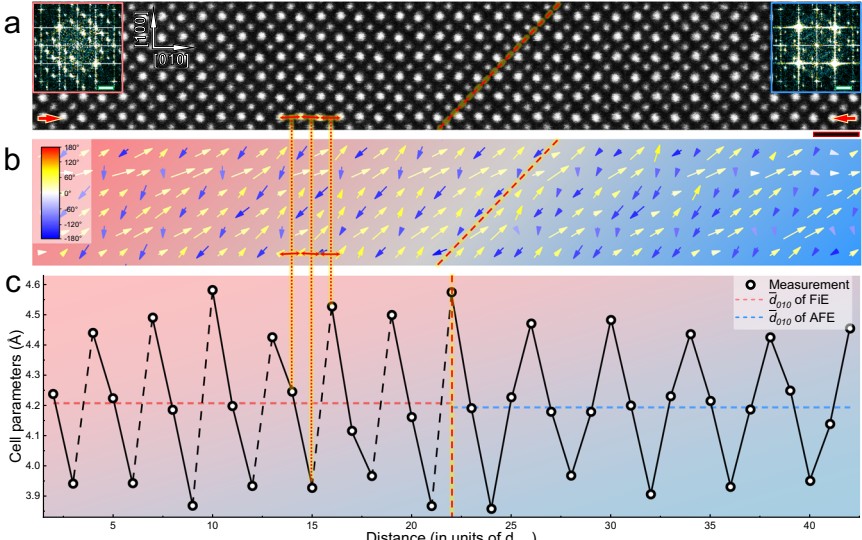

**Fig. 4 | Atomic-scale characterizations of phase separation in PZ. a** [001]-projected HAADF image of (left) FiE-like and (right) AFE-like phase at the phase boundary region. The scale bar refers to 1 nm. The insets show Fourier transform images of two-phase regions with the scale bars of 2 1/nm. **b** Displacement mapping of Pb cations (see arrows), with respect to the atomic image in **a**. **c** The variation of atomic spacings (represented by circles) along the pair of red arrows in **a**. The mean values of $d_{010}$ for FiE- and AFE-like phases are indicated by a light red and blue horizontal dashed line, respectively. The phase boundary is marked by an oblique red dashed line in **a** and **b**, a vertical red dashed line in **c**, respectively.

grows to the observable size, leading to the coexistence of both FiE-like and AFE-like phase structures (Fig. 5c).

Generally, the thicker PZ films ought to exhibit more AFE-like phase because the compressive strain initiated from the substrate could be more relaxed with thickness. However, on the contrary, the FiE-like phase increased its amount with the increment of film thickness as seen in Fig. 2. After fitting the peaks in the intensity profiles and integrating the curves of two groups of peaks, we quantified the volume fraction of the FiE-like phase as illustrated in Fig. S12. A high-volume fraction of more than 80 vol% is thus obtained in the 180 nm thick film, i.e., even higher amount than the AFE-like phase volume.

The evolution of volume fraction of FiE phase with film thickness could be correlated to the construction of APBs. As most of the misfit strain could be relaxed by the misfit dislocations at the semi-coherent interfaces, it is generally considered that the additional strain resulted from thermal coefficient mismatch and paraelectric-antiferroelectric phase transition during cooling is still large and requires further release via other ways, e.g. phase separation and the formation of APBs. With the pre-existed FiE-like embryos at APBs, the FiE-like phase tends to expand from APBs and finally is stabilized at equilibrium state coexisted with AFE phase. It has been reported that the type of translational boundaries could be related to the local strain induced by the dislocation cores[27]. Besides, the film thickness and thermal coefficient mismatch may also contribute to the density of translational boundaries. The number of "↑↑↓"-like APBs in a thicker film is supposed to be smaller than the one in a thinner film attributed to larger strain relaxation in thicker films. Thereafter, fewer APBs would consequently reduce the clamping effect on the FiE phase from the neighboring AFE phases at both sides and provide more room for the quick growth of the FiE phase during cooling process as seen in Fig. 2 and Fig. S6. In the future, different substrates could be used to change the degree of compressive strain or even provide the tensile strain to modulate the interfacial defects like APBs.

To summarize, the theoretically predicted low-temperature threefold FiE-like state is experimentally obtained at room temperature through strain-mediated phase separation in PZ/STO films. The FiE-like phase manifests itself in a nanoscale lamellar dual-phase (AFE and FiE) intergrowth construction and shows a threefold modulation period. Moreover, the volume fraction of FiE-like phase unexpectedly increases with film thickness. Pb cations displacement mapping demonstrates approximate "↑↑↓" configuration in FiE-like phase and "↑↑↓↓" configuration in AFE-like phase, which are consistent with the prediction of physical model in a recent theoretical study but show comparatively larger disturbance of polar dipoles. GPA analysis reveals that most of the compressive strain in PZ film induced by the STO substrate could be relaxed by periodic misfit dislocations at the interface. The misfit dislocations along with thermal coefficient mismatch and paraelectric-antiferroelectric phase transition during cooling finally determine the type and formation of APBs which consequently initiate the film-thickness tailored phase separation. Finally, nanoscale lamellar dual-phases (AFE and FiE) are generated and stabilized in PbZrO₃ thin films and the volume fraction of the FiE phase increases with film thickness.

In terms of phase manipulation, the epitaxial strain has been found to have wide effects on phase transition in the perovskite films, manganite films and other related systems. For instance, a tetragonal (T)-like phase in BiFeO₃ with larger $c/a$ ratio can be stabilized by epitaxial strain[28] while the common state of BiFeO₃ used to be considered as rhombohedral (R) phase. In a manganite film, the antiferromagnetic metallic ground state can be transformed to a ferromagnetic phase by a uniaxial tensile (or compressive) strain[29,30]. These cases and our work demonstrate that the epitaxial strain driven phase transition could be a universal phenomenon. The present achievement of stable strongly competing phases at room temperature fertilizes the understanding of diverse configurational tailoring of dipoles ordering in PbZrO₃ and evokes further study on the physical properties of FiE phase in the future.

## Methods
### Fabrication and electrical properties of thin films
The PZ films were fabricated by chemical solution deposition method. Firstly, the raw materials of lead(II) acetate trihydrate (C₄H₆O₄Pb·3H₂O, AR, Sinopharm Chemical Reagent Co. Ltd.) and acetylacetone (AR, Sinopharm Chemical Reagent Co. Ltd.) were dissolved in acetic acid (AR, Sinopharm Chemical Reagent Co. Ltd.), with 20 mol% excess lead acetate added to compensate the

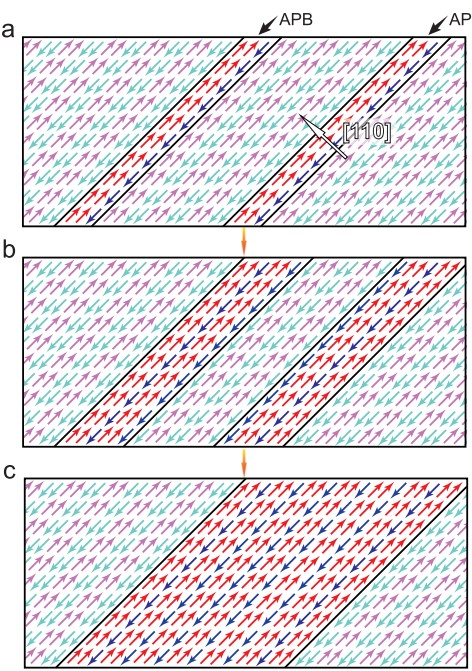

**Fig. 5 | The schematic diagrams of nucleation and growth of FiE-like phase.**
**a** The "↑↑↓" FiE-like configuration of APBs. **b** The coarsened APBs and the embryo of the FiE phase. **c** The generation of the stripe-like mixed-phases.

lead volatilization loss. Secondly, zirconium (IV) n-propoxide ($Zr(OCH_2CH_2CH_3)_4$, 70 wt.% in propanol, Aldrich) was added into the solution and kept the solution stirring for 1 h. Thirdly, the precursor solution was spin-coated on $SrTiO_3$ substrates (0.7 wt% Nb doping $SrTiO_3$ substrates for electrical tests) (HF-Kejing Material Technology Co., Ltd.) and the films were pyrolyzed at 350 °C and at 600 °C respectively. These steps were repeated to obtain the desired thickness. Finally, the films were annealed at 700 °C in a rapid thermal processing furnace. Au top electrodes with diameters of ~250 μm were deposited on the film surface via stainless steel shadow mask. The P-E hysteresis loops and dielectric tunability were conducted by using a commercial ferroelectric analyzer (TF Analyzer 3000, aixACCT, Aachen, Germany).

### TEM sample preparation and data acquisition
The samples for transmission electron microscopy were prepared by focused ion beam (Helios 5 UX, FEI), using an Ar ion beam first at the voltage of 30 kV and the current of 2.4 nA and then, at the voltage of 2 kV and the current of 50 pA to avoid strong ion-beam damage. The dark-field (DF) images and selected-area electron diffractions (SAED) were acquired on a transmission electron microscope (JEM-2100F, JEOL, Japan) at 200 kV. The atomic-scale STEM HAADF images were acquired on a probe corrected transmission electron microscope (HF5000, Hitachi, Japan, at 200 kV). The convergence semi-angle is 20 mrad. The probe size is in UHR mode and the collection semi-angle is 60–320 mrad.

### Geometric phase analysis (GPA)
The GPA was performed using Strain++ program (developed by Jonathan Peters, https://github.com/JJPPeters/Strainpp) based on Martin Hÿtch's reported algorithm[31]. The superlattice spots were chosen to calculate the local strain field in PZ, and the center of Gaussian mask corresponds to the region between T- and Q-phase, which serves as the reference lattice (corresponding Fourier transform of images are exhibited in Fig. S13).

### First-principles density functional theory (DFT) calculations
The DFT calculations were performed using the Vienna ab initio simulation package (VASP)[32]. The projector augmented wave (PAW) method was used for describing electron-ion interactions[33] and the generalized gradient approximation (GGA) parametrized by Perdew-Burke-Ernzerhof for solids (PBEsol) was used for the exchange-correlation functionals[34]. To verify the robustness of the DFT results regarding the choice of exchange-correlation functionals, we also compare the energy vs. strain curves computed with three other functionals, viz. Perdew-Burke-Ernzerhof (PBE)[35], local-density approximation (LDA), and strongly constrained and appropriately normed (SCAN) meta-GGA[36] (see Fig. S14). The orbitals of $5d^{10}6s^26p^2$, $4s^24p^64d^25s^2$, and $2s^22p^4$ were explicitly treated as valence electrons for Pb, Zr, and O, respectively. A cutoff energy of 500 eV for the plane-wave basis set and $\Gamma$-centered $6 \times 3 \times 8$ and $8 \times 4 \times 6$ **k**-point mesh were used for the FiE (60 atoms, space group: *Ima2*) and *Pbam* (40 atoms) phase[37], respectively. All atoms were relaxed until the Hellmann-Feynman forces were less than 0.001 eV/Å. To be consistent with the film orientation in experiment, (001) epitaxial biaxial strain is applied to *Pbam* and FiE phases, with which the space groups of both phases preserve under strained conditions. Meanwhile, for direct comparison with experiments, the cube root of the volume per f.u. in the ground state of *Pbam*: $a_0 = 4.135$ Å with PBEsol ($a_0 = 4.186$ Å with PBE, $a_0 = 4.100$ Å with LDA, and $a_0 = 4.151$ Å with SCAN) was chosen as the initial lattice constant to calculate the epitaxial strain.

### The atom position measurement
The picometer-precision fitting of atomic columns was done by using StatSTEM (MATLAB code), a least squares estimation algorithm for accurate and precise quantification of the atomic column positions and intensities from atomic-scale images with considering overlap between neighboring atomic columns, which was developed by de Backer et al.[38] and has been widely used in many published works[8,39].

Although the previous study[40] indicated that the configuration of Pb displacements can be determined without observable artifacts even in the extreme case of an 80 nm thick $PbZrO_3$ sample at tilting of 5 mrad in both α and β directions, we still tried our best to correct mistilt and aberrations during experiments. Furthermore, we repeatedly and rapidly scan the same area and then cross correlate and sum all the single-frame images to decrease drift effect.

Generally, there are two different methods that can be used to calculate the Pb displacements of $PbZrO_3$-based materials: (1) by referring to the averaged surrounding Zr positions[41], (2) by averaging the lattice of Pb atom[42]. The first method was chosen in the present study because Pb atoms have obviously larger displacement than B-site cations in $PbZrO_3$-based materials[42,43].

## Data availability
The data that support the findings of this study are available from the corresponding author upon reasonable request.

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

## Acknowledgements

This work was supported by National Natural Science Foundation of China (U2230104, 52002388 and U2002217), Young Elite Scientists

Sponsorship Program by CAST (2022QNRC001), Shanghai Rising-Star Program (23QA1410800), Shanghai Science and Technology Innovation Action Plan (21ZR1472400) and Shanghai Technical Platform for Testing and Characterization on Inorganic Materials (19DZ2290700). Particularly, N.F. and B.X. acknowledge the financial support from National Natural Science Foundation of China (Grant No. 12074277), Projects of International Cooperation and Exchanges NSFC (Grant No. 12311530693), Jiangsu Shuangchuang Project (JSSCTD202209) and Natural Science Foundation of Jiangsu Province (BK20201404).

## Author contributions

Z. Y., Z. F. and F. X. conceived the project. Z. Y., L. Z. and Y. Z. performed the imaging experiments and analyzed the imaging data. B. H, S. Y, H. C. and X. C. helped in the synthesis of film materials. N. F. and B. X. performed the simulations. Z. Y., Z. F. and F.X. drafted the paper. Z. F., F. X. and G. W. supervised the research. All authors discussed the results and commented on the paper.

## Competing interests

The authors declare no competing interests.
