## [Peer Review File · Nature Communications]

Room-temperature Stabilizing Strongly Competing
Ferrielectric and Antiferroelectric phases in PbZrO₃ by Strain
Mediated Phase SeparationEditorial Note: Parts of this Peer Review File have been redacted as indicated to remove third-party material where no permission to publish could be obtained.

REVIEWER COMMENTS

Reviewer #1 (Remarks to the Author):

In a recent study, researchers challenged the conventional understanding of PbZrO₃ as an antiferroelectric material by demonstrating that it can exist in a novel ferrielectric phase, typically accessible at low temperatures. They successfully achieved room-temperature coexistence of the ferrielectric and antiferroelectric states in PbZrO₃ thin films by using strain-mediated phase separation on a SrTiO₃ substrate. This breakthrough not only provides experimental evidence for the existence of a room-temperature ferrielectric phase but also offers opportunities for exploring its unique properties and potential applications.

The work is interesting and present innovation in the field, however there are several issues needed to be improved before the publication in this high quality journal. Remarks are listed.

1. The XRD presented in the supplementary data revealed that PbZrO₃ remained very weak compared to SrTiO₃ substrate. From my view point, this often happens when the quality of the film is not good and the surface exhibits a porous structure. The Author should you elucidate this point by providing high quality SEM images.

2. What factors influence the stability and preference for either the conventional antiferroelectric (AFE) phase or the theoretically predicted ferrielectric (FiE) phase in PbZrO₃ thin films, particularly when considering strain within the pseudo-cubic (100) and (001) planes?

3. What are the key factors and mechanisms contributing to the stabilization and coexistence of FiE-like and AFE-like phases in PbZrO₃ thin films, especially with regard to their sensitivity to film thickness, compressive strain, and the presence of interfacial defects, such as antiphase boundaries (APBs)?

4. The referee suggests enriching the discussion based on the experimental results, which will be very important for the readers in the relative field. More recent literature is suggested to be included.

In general, this work seems to be interesting and the referee would like to see the revision.

Reviewer #2 (Remarks to the Author):

PZO is a classic AFE material which shows more and more interesting microstructures and related responses. While it is 'classic', we have lots of unknowns on this perovskite oxide, such as its FE-AFE phase transition details and related properties. These mysteries stop us knowing about the so-called antiferroelectric material, and thus probably impedes us developing high-power energy storage materials. Recently, a special ferrielectric phase has been found in PZO, which was predicted in 2011 and 2022. Importantly, this ferrielectric phase was successfully observed by Y. Liu et. al, at PHYSICAL REVIEW LETTERS 130, 216801 (2023) and R. J. Jiang et. al., Nano Lett. 2023, 23, 1522–1529, respectively. These achievements remind us of the fact that we do not really know the classic AFE PZO quite well. Here the authors have obtained even 80 vol% ferrielectric phase at room temperature for the strain mediated PZO films grown by chemical solution deposition. In particular, the mixed phases here show special domain structures, which is a big step in tuning the PZO microstructures and related properties. I believe this is important for future exploration of

structures, physical properties and applications based on the novel ferrielectric phase I have some concerns which need to be addressed:

First about the GPA and strains. The author says 'It can be clearly seen that the domain configuration exhibits a typical a/c multidomain for minimizing the elastic energy.'. Note that here even the smaller a lattice of PTO is much larger than LAO, thus the a domains here do not 'minimize the elastic energy' at all, they tend to increase the elastic energy by only considering the misfit. The origin of a domains thus must be induced by other reasons. More discussions are needed to interpret this issue.

Moreover, questions arise from Fig. 3e-3h, . Please note that for a given digital image, the strains extracted from this image should be the same, no matter what kinds of methods were used. Please note that in Fig. S10c, there is no sense by using $1/4\langle 130 \rangle$ and $1/3\langle 120 \rangle$, $1/4\langle 310 \rangle$ and $1/3\langle 210 \rangle$, to extract strains simultaneously from both AFE and FiE phases, since the $1/4$ spots only contain AFE lattice information and the $1/3$ spots only contain FiE lattice information. Here, for extracting the strain maps from high-resolution TEM images, a precondition is to choose a lattice frame which must contain all phases. By the way, HAADF-STEM images based on aberration corrected STEM are more suitable for doing GPA.

Then, the claim 'It seems that PS rather than the domain boundaries is responsible for the relaxation of the strain arisen from a lattice misfit as large as -6.5%.' is not true based on the discussion above. In Fig. S2a, it is clear that misfit dislocations are responsible for most relaxations of the PZO/STO system.

In summary, this is an important observation indicating that chemical-based method may be effective for tuning both structures and properties of FiE phase in PZO. The author need pay special attention on the interpretations of strains and misfit relaxations, and strain engineering etc..

Reviewer #3 (Remarks to the Author):

It is unfortunate that the authors have neglected important scientific advances in this field and present their "important contribution" in a somewhat misleading manner. The authors claim that this is the "first experimental evidence" for the observation of a ferrielectric phase in PbZrO₃, but this is not true.

A ferrielectric phase has already been reported in PbZrO₃ thin films grown on Si substrates (Ferrielectricity in the Archetypal Antiferroelectric, PbZrO₃, *Advanced Materials*, 2023, 35, 2206541). This impairs the novelty and importance of this manuscript. In addition, a ferrielectric-like structure was also reported in PbZrO₃/SrRuO₃/SrTiO₃ thin films (reference 19).

If one reads reference 18 carefully (On the possibility that PbZrO₃ not be antiferroelectric), it is also not true that the ferrielectric phase is accessible only up to 255K, as stated in the abstract. Also, we should note that there is no solid theoretical understanding towards ferrielectricity or antiferroelectricity so far, especially since there are conflicting viewpoints. For example, Baker et al. suggested an 80-atom Pnam phase as the ground state of PbZrO₃ (A re-examination of antiferroelectric PbZrO₃ and PbHfO₃: an 80-atom Pnam structure, arXiv:2102.08856). In sight of the above concerns, the motivation in the abstract and introduction sections are not valid.

Some other comments:

a) Important electrical characterizations for antiferroelectric studies, such as double PE loops and dielectric tunability, are missing from this manuscript but they are well covered in the previous work (Advanced Materials, 2023, 35, 2206541).

b) If there were two phases coexisting in the materials, it would be interesting to know if there are differences in the chemical compositions.

c) Standard θ - 2θ scans are not sufficient to demonstrate good epitaxial growth. Additional structural characterizations are required. In particular, reciprocal space mapping should be done to determine the actual lattice parameters of PbZrO₃ films.

d) It appears that the FiE phase is more stable at zero strain in Figure 1A.

e) The streaking features observed for the diffraction patterns are not explained, and they are least visible in the 180-nm sample. Such features are not reflected in the modelled patterns.

Response Letter

Manuscript ID: NCOMMS-23-48001

The authors sincerely thank the reviewers and the editor for the constructive comments and suggestions which help to significantly improve the quality of the manuscript. We have spent much time and efforts doing supplementary work to properly respond to the reviewers' questions. The manuscript has now been carefully revised accordingly. The point-by-point responses are listed below and the corresponding revisions have been highlighted in yellow in the revised manuscript.

Responses to Reviewer #1

Comment 1:

In a recent study, researchers challenged the conventional understanding of PbZrO_3 as an antiferroelectric material by demonstrating that it can exist in a novel ferrielectric phase, typically accessible at low temperatures. They successfully achieved room-temperature coexistence of the ferrielectric and antiferroelectric states in PbZrO_3 thin films by using strain-mediated phase separation on a SrTiO_3 substrate. This breakthrough not only provides experimental evidence for the existence of a room-temperature ferrielectric phase but also offers opportunities for exploring its unique properties and potential applications. The work is interesting and present innovation in the field, however there are several issues needed to be improved before the publication in this high-quality journal. Remarks are listed.

Response:

We sincerely thank the reviewer for highlighting the importance of our study, and for the following constructive comments, which strongly encourage us to further improve the quality of the manuscript. We carefully go through the reviewer's remarks and give our answers below.

Comment 2:

The XRD presented in the supplementary data revealed that PbZrO_3 remained very weak compared to SrTiO_3 substrate. From my view point, this often happens when the quality of the film is not good and the surface exhibits a porous structure. The Author should you elucidate this point by providing high quality SEM images.

Response:

The authors highly appreciate the reviewer's professional comment and suggestion. The cross-sectional SEM images are presented in Fig. R1. As the reviewer pointed out, all three films exhibit a porous structure. Fig. R1 is also presented in supporting information as Fig. S3 and the corresponding description has been added in the revised manuscript on page 4.

Fig. R1. Cross-sectional SEM images of PbZrO₃/SrTiO₃ thin films with film thickness of (a) ~60, (b) ~160 and (c) ~180 nm.

Comment 3:

What factors influence the stability and preference for either the conventional antiferroelectric (AFE) phase or the theoretically predicted ferrielectric (FiE) phase in PbZrO₃ thin films, particularly when considering strain within the pseudo-cubic (100) and (001) planes?

Response:

Owing to the small free energy discrepancy, the final manifested phase state in PZO can be perturbed by external electric field, defects, electrostatic boundary conditions, chemical compositions, film thickness, growth orientations and temperatures, and so on⁽¹⁻⁵⁾. For instance, the variation of the chemical composition (the doping effect of B-site Sn/Ti elements) in Pb_{0.97}La_{0.02}(Zr_{0.50}Sn_xTi_{0.50-x})O₃ ceramics could induce the modulation period to change from fourfold (antiferroelectric phase) to sixfold or ninefold (ferrielectric phase)⁽³⁾ (our work: *Nat. Commun.*, 2020, 11, 3809). However, the theoretically predicted threefold period ferrielectric phase has not been successfully achieved by all these means. In this work, the threefold ferrielectric phase was successfully induced by compressive stress.

According to the DFT calculation in Fig. 1, the AFE and FiE phases demonstrate different preferential growth behaviors that the AFE phase is more favorable under (100) plane biaxial compressive strain, while the FiE phase is more favorable under (001)

strain. The experimental observations indicate that the phase of PZO is neither pure AFE nor pure FiE, but the coexistence of both via an alternating arrangement. This phenomenon implies that both the strain from substrate and the interphase strain contribute to the final total free energy.

In addition, the effect of strain is dependent on the configuration of polarization. Despite both AFE and FiE have the polarization along $\pm[1\bar{1}0]$ direction on the (001) plane, the compressive stress may have different effects on the stability of “ $\downarrow\downarrow\uparrow\uparrow$ ” configuration or “ $\downarrow\downarrow$ ” configuration depending on the direction of stress with respect to the polarization direction. It seems that a compressive stress with its direction out of the polarization plane would favor the stability of the FiE phase (see Fig. 1b). However, the exact reason for the occurrence of only the (001) growth in the practical film preparation requires further study in the near future. The enriched discussion could be found on page 4 (just above Fig. 1) in the revised manuscript.

Comment 4

What are the key factors and mechanisms contributing to the stabilization and coexistence of FiE-like and AFE-like phases in PbZrO_3 thin films, especially with regard to their sensitivity to film thickness, compressive strain, and the presence of interfacial defects, such as antiphase boundaries (APBs)?

Response:

According to the experimental results that all three samples exhibit coexistence of FiE-like and AFE-like phases while the thicker sample show larger amount of FiE-like phase, it can be concluded that high enough compressive strain is a fundamental premise for inducing FiE-like phase and the antiphase boundaries are the key factor for tuning the relative amount of FiE-like and AFE-like phases. As predicted in the reported theoretical work, the compressive strain can facilitate the FiE-like phase to overcome energy barrier and manifest itself against AFE-like phase. Thus, we selected STO as a substrate to provide compressive strain environment in PZO film.

On this basis, thicker films are supposed to release more compressive stress and thus should decrease the amount of FiE-like phase. However, we observed that the volume fraction of FiE phase is increased in thicker films. In our opinion, this phenomenon is most likely due to that the number of APBs may have decreased in a thicker film. During the formation of modulated structures, many interfacial defects, i.e. APBs will spontaneously occur. These APBs usually exhibit “*up-up-down*”-like threefold ferrielectric structure ^{(4), (5)}, which might act as intrinsic nucleation sites for the generation of the FiE-like phase. Meanwhile, according to the observed interfacial defect driven phase transition in PbZrO_3 -based materials ⁽⁶⁾, it is reasonable to speculate that the pre-existed FiE-like phase at the APBs could further grow under applied external stress. As shown in the schematic diagrams in Fig. R2a, the APBs exhibit one unit of *up-up-down* FiE-like configuration. During the preparation of thin film (especially at the cooling stage under compressive stress), the APBs are coarsened and the embryonic form of the FiE phase is generated (Fig. R2b). Finally, the FiE-like phase further grows to the observable size (Fig. R2c), leading to the coexistence of both FiE-

like and AFE-like phase structures.

Thus, the decreased number of APBs in thicker films would consequently reduce the clamping effect on the FiE phase from the neighboring AFE phases at both sides and provide more room for the quick growth of the FiE phase. The Fig.R2 was presented as Fig. S8 in SI in the initial submission and now is shown in the main text of the revised manuscript as Fig. 5 due to its importance, and the corresponding description is added in the text on page 9 and 10 (please also see response to the comment 5 below).

Fig. R2. The schematic diagrams of nucleation and growth of FiE-like phase. The APBs locate on the (110) plane.

Comment 5

The referee suggests enriching the discussion based on the experimental results, which will be very important for the readers in the relative field. More recent literature is suggested to be included.

Response:

We accept the reviewer's suggestion. The enriched discussion has been added and some recent literatures have been included in the revised manuscript (see page 9 and 10). The discussion is also presented here for the reviewer's convenience.

“The evolution of volume fraction of FiE phase with film thickness could be correlated to the construction of APBs. As most of the misfit strain could be relaxed

by the misfit dislocations at the semi-coherent interfaces, it is generally considered that the additional strain resulted from thermal coefficient mismatch and paraelectric-antiferroelectric phase transition during cooling is still large and requires further release via other ways, e.g. phase separation and interfacial defects, i.e. APBs. With the pre-existed FiE-like embryos at APBs, the FiE-like phase tends to expand from APBs and finally is stabilized at equilibrium state coexisted with AFE phase. It has been reported that the type of translational boundaries could be related to the local strain induced by the dislocation cores ⁽⁷⁾ (*Adv. Mater., Interfaces* 2015, **2**, 1500349). Besides, the film thickness and thermal coefficient mismatch may also contribute to the density of translational boundaries. The number of “↑↑↓”-like APBs in a thicker film is supposed to be smaller than the one in a thinner film attributed to larger strain relaxation in thicker films. Thereafter, fewer APBs would consequently reduce the clamping effect on the FiE phase from the neighboring AFE phases at both sides and provide more room for the quick growth of the FiE phase during cooling process as seen in Fig. 2 and Fig. S6. In the future, different substrates could be used to change the degree of compressive strain or even provide the tensile strain to modulate the interfacial defects like APBs.”

Comment6

In general, this work seems to be interesting and the referee would like to see the revision.

Response:

We thank the reviewer again for the precious comments and suggestion.

Responses to Reviewer #2

PZO is a classic AFE material which shows more and more interesting microstructures and related responses. While it is ‘classic’, we have lots of unknowns on this perovskite oxide, such as its FE-AFE phase transition details and related properties. These mysteries stop us knowing about the so-called antiferroelectric material, and thus probably impedes us developing high-power energy storage materials. Recently, a special ferrielectric phase has been found in PZO, which was predicted in 2011 and 2022. Importantly, this ferrielectric phase was successfully observed by Y. Liu et. al, at PHYSICAL REVIEW LETTERS 130, 216801 (2023) and R. J. Jiang et. al., Nano Lett. 2023, 23, 1522–1529, respectively. These achievements remind us of the fact that we do not really know the classic AFE PZO quite well. Here the authors have obtained even 80 vol% ferrielectric phases at room temperature for the strain mediated PZO films grown by chemical solution deposition. In particular, the mixed phases here show special domain structures, which is a big step in tuning the PZO microstructures and related properties. I believe this is important for future exploration of structures, physical properties and applications based on the novel ferrielectric phase

I have some concerns which need to be addressed:

Response:

We sincerely thank the reviewer for the positive comments on our study, and for constructive suggestions with respect to the content of the manuscript. It should be noted that the predicted strongly competing FiE phase against AFE ground state refer to the specific dipole configuration of *up-up-down* with threefold modulated period. In contrast, the most reported FiE phases exhibit a modulated period larger than fourfold. However, as the reviewer mentioned, the observations of threefold FiE phase in “*Nano Lett., 2023, 23, 1522–1529*”⁽⁸⁾ and “*Phys. Rev. Lett., 2023, 130, 216801*”⁽⁹⁾ undoubtedly demonstrate that this predicted FiE phase can be achieved at room temperature despite only focused on translational boundaries and phase transition. Here, we not only achieved a specific pattern of alternating phase coexistence of FiE and AFE phase, but also manipulated the volume fraction of FiE phase by changing the film thickness. We highly appreciate the reviewer’s profound knowledge and the professional comments. Next, we carefully go through reviewer’s remarks and give our answers below.

Comment 1:

First about the GPA and strains. The author says ‘It can be clearly seen that the domain configuration exhibits a typical *a/c* multidomain for minimizing the elastic energy.’. Note that here even the smaller *a* lattice of PTO is much larger than LAO, thus the domains here do not ‘minimize the elastic energy’ at all, they tend to increase the elastic energy by only considering the misfit. The origin of *a* domains thus must be induced by other reasons. More discussions are needed to interpret this issue.

Moreover, questions arise from Fig. 3e-3h. Please note that for a given digital image, the strains extracted from this image should be the same, no matter what kinds

of methods were used. Please note that in Fig. S10c, there is no sense by using $1/4\langle 130 \rangle$ and $1/3\langle 120 \rangle$, $1/4\langle 310 \rangle$ and $1/3\langle 210 \rangle$, to extract strains simultaneously from both AFE and FiE phases, since the $1/4$ spots only contain AFE lattice information and the $1/3$ spots only contain FiE lattice information. Here, for extracting the strain maps from high-resolution TEM images, a precondition is to choose a lattice frame which must contain all phases. By the way, HAADF-STEM images based on aberration corrected STEM are more suitable for doing GPA.

Response:

We sincerely thank for the reviewer's comments and suggestions. Firstly, we agree with the reviewer's comments with respect to a/c multidomain in PTO film. The formation of a/c multidomain is not for minimizing the elastic energy but for decreasing the total free energy including elastic, electrostatic, and domain-wall energies⁽¹⁰⁾. In this case, the content with respect to PbTiO_3 doesn't make much sense to understand the observed phase separation in PZO film and has been deleted in the revised manuscript.

Secondly, we accept the reviewer's suggestion on GPA in PZO film and acquire HAADF-STEM images to reperform GPA, as shown in Fig. R3 (also is the revised Fig. 3). We captured the atom-resolved HAADF-STEM image as shown in Fig. R3a, and the corresponding GPA analysis using the $(0\bar{1}0)$ and (100) basic reflections are presented in Fig. R3b (ϵ_{xx}) and Fig. R3c (ϵ_{yy}). It can be seen that the compressive strain from the substrate is almost totally relaxed via forming misfit dislocations and there is no obvious stress gradient throughout the thin films. The uniform strain state suggests the approximately equal a/b lattice parameters of AFE- and FiE-like phases, which matches well with the new supplementary RSM results in Fig. R6 (please see the response to the reviewer 3, comment 4-c).

Although the reviewer pointed out that there is no sense to characterize the local strain by selecting satellite reflections to do the GPA, we found that it is a convenient way to visualize the distribution of three and fourfold phases on the HAADF images (Fig. R3e and Fig. R3f), which in the meantime let us easily observe their correspondent sites with the misfit dislocations at the film interface.

In Fig. R3b, the sites of stress concentration along the in-plane direction at the PZO/STO interface refer to the misfit dislocation cores, which should not directly contribute to the stabilization and coexistence of FiE-like and AFE-like phases in PbZrO_3 thin films. According to Fig. R3b and Fig. R3e, the GPA images demonstrate that both of FiE phase and AFE regions cover the misfit dislocations, implying that there is no direct correspondence between the nucleation sites of FiE (or AFE) phase and the core of misfit dislocation. However, the local strain induced by these dislocations can influence the formation of APBs⁽⁵⁾, which can indirectly intervene the stabilization and coexistence of FiE-like and AFE-like phases (Please also see response to comment 4 of reviewer 1).

The corresponding data and discussion (see page 6 and 7) have been added in the revised manuscript and the Fig. 3 is updated.

Fig. R3. Strain fields in 180nm PbZrO₃ thin film via GPA analysis. (a) HAADF-STEM image of the cross-sectional PZ/STO thin film. (b, c) The corresponding maps of local strain fields compared with the STO substrate. (e, f) The corresponding maps of distribution of FiE and AFE phase. (d) The Fourier transform of image (a). The yellow circles in (d) mark the Gaussian masks for (b, c) and the red ones for (e, f).

Comment 2:

Then, the claim ‘It seems that PS rather than the domain boundaries is responsible for the relaxation of the strain arisen from a lattice misfit as large as -6.5%.’ is not true based on the discussion above. In Fig. S2a, it is clear that misfit dislocations are responsible for most relaxations of the PZO/STO system.

Response:

The authors highly appreciate the reviewer’s professional comment and prudent consideration. Also, we thank for the kind suggestion of using HAADF-STEM images to do GPA, which could demonstrate the periodic misfit dislocations clearly. As pointed out by the reviewer, the GPA based on HAADF-STEM images indeed indicates that the misfit strain is mostly released by the misfit dislocations. The corresponding description has been added in the revised manuscript (see page 6 and 7).

Comment 3:

In summary, this is an important observation indicating that chemical-based method may be effective for tuning both structures and properties of FiE phase in PZO. The author needs pay special attention on the interpretations of strains and misfit relaxations, and strain engineering etc.

Response:

We highly appreciate the reviewer’s constructive comments and suggestion. According to the reviewer’s suggestion, we now get more insights between compressive strains and misfit relaxations by analyzing the atomic HAADF-STEM images. In the future, we may select different substrates to change the degree of lattice misfit to further investigate how the AFE and FiE phase evolve with misfit dislocations and compressive strain. Especially, a MgO substrate can be used because it is an only substrate that can provide tension strain. Again, we thank the reviewer very much for giving us so many professional suggestions for helping us to improve the quality of this work.

Responses to Reviewer #3

Comment 1:

It is unfortunate that the authors have neglected important scientific advances in this field and present their “important contribution” in a somewhat misleading manner. The authors claim that this is the “first experimental evidence” for the observation of a ferrielectric phase in PbZrO_3 , but this is not true.

A ferrielectric phase has already been reported in PbZrO_3 thin films grown on Si substrates (Ferrielectricity in the Archetypal Antiferroelectric, PbZrO_3 , *Advanced Materials*, 2023, 35, 2206541). This impairs the novelty and importance of this manuscript. In addition, a ferrielectric-like structure was also reported in $\text{PbZrO}_3/\text{SrRuO}_3/\text{SrTiO}_3$ thin films (reference 19).

Response:

The reviewer pointed out some publications on ferrielectric phases. However, here we would like to emphasize that the predicted strongly competing FiE phase against AFE ground state refer to the specific dipole configuration of *up-up-down* with threefold modulation period ⁽¹²⁾ (*npj Comput. Mater.*, 2021, 7, 196). In contrast, so far most reported FiE phases including the one the reviewer mentioned exhibit a modulated period larger than fourfold. For example, the FiE phase found in PbZrO_3/Si film has ultra-long-period with configuration of local ferroelectric phase ⁽¹³⁾ (*Adv. Mater.*, 2023, 35, 2206541) (shown in Fig. R4 for reviewer’s convenience) and the one reported in $\text{PbZrO}_3/\text{SrRuO}_3/\text{SrTiO}_3$ thin films shows the “↑↑↑↑ ↓↑↑↓” configuration ⁽¹⁴⁾ (reference 19, *Phys. Rev. B*, 2022, 105, 125409). These different phases can be observed because they also exhibit similar energy with archetypal *Pbam* AFE phase according to the calculated results of PZO. Here, we not only observed FiE phase with *threefold* modulation period, but also achieve a relative larger vol% of FiE phase in the pattern of phase coexistence of alternating FiE and AFE phase. In order to clarify the difference in ferrielectric configurations between the available publications and this work, we complement certain description in the second paragraph of the revised manuscript. We highly appreciate the reviewer’s professional comments and suggestions. Next, we carefully go through reviewer’s remarks and give our answers below.

[Redacted]

Fig. R4. The polarization configurations from Pb ions displacements in 001- (a) and 042-oriented (b) film in the report “*Adv. Mater.*, 2023, 35, 2206541”.

Comment 2:

If one reads reference 18 carefully (On the possibility that PbZrO_3 not be antiferroelectric), it is also not true that the ferrielectric phase is accessible only up to 255K, as stated in the abstract. Also, we should note that there is no solid theoretical understanding towards ferrielectricity or antiferroelectricity so far, especially since there are conflicting viewpoints. For example, Baker et al. suggested an 80-atom Pnam phase as the ground state of PbZrO_3 (A re-examination of antiferroelectric PbZrO_3 and PbHfO_3 : an 80-atom Pnam structure, arXiv:2102.08856). In sight of the above concerns, the motivation in the abstract and introduction sections are not valid.

Response:

We highly appreciate the reviewer's rigorous scholarship. In case there is no solid theoretical understanding towards ferrielectricity or antiferroelectricity so far and there are conflicting viewpoints, the experimental confirmation of these theoretically predicted phase in PZO will facilitate further understanding towards PZO. The theoretical works indicate that there are several phases exhibiting similar energy with archetypal *Pbam* AFE phase, such as the 30-atom *Ima2* FiE phase and the 80-atom *Pnam* AFE-like phase. Although these predicted phases exhibit very small energy difference with the *Pbam* AFE phase, the *Ima2* FiE phase is a special one because it has a surprisingly threefold modulation period while the most reported FiE phases have larger modulation period than the fourfold archetypal *Pbam* AFE phase ^{(6), (13)}. In the article of reference 18, the calculated results reveal the threefold FiE state to be more stable than the commonly accepted antiferroelectric phase at low temperatures, or possibly up to room temperature. Recently, the observations in PZO single crystal and film undoubtedly demonstrate that the predicted threefold FiE phase can be achieved at room temperature despite only focused on translational boundaries and phase transition ^{(8), (9)}. However, it is unclear that how the specific threefold FiE coexists with AFE phase and whether the threefold FiE phase could be manipulated at ambient temperatures. Based on the above-mentioned progresses, we accept the reviewer's suggestion and revised the corresponding introductions in the revised manuscript. (page 2)

Comment 4:

Some other comments:

a) Important electrical characterizations for antiferroelectric studies, such as double PE loops and dielectric tunability, are missing from this manuscript but they are well covered in the previous work (*Advanced Materials*, 2023, 35, 2206541).

Response:

We appreciate and accept the reviewer's suggestions. By using the Nb doped SrTiO₃ substrates as the bottom electrode and Au as the top electrode, the measured double P-E loops for 60, 160 and 180 nm thin films are shown in Fig. R5a~R5c (also presented as Fig. S4 in SI). The evolution of P-E hysteresis loops (Fig. S4) demonstrates gradual occurrence of bulging shape near the zero-electric field upon increasing the film thickness. This bulging shape indicates increased remnant polarization in thicker film, which is most likely related to the presence of mixed-phases of AFE and FiE state. The curves of dielectric permittivity-DC electric fields in 60, 160 and 180 nm thin films are shown respectively in Fig. R5d~f. We add the P-E hysteresis loops and the curves of dielectric permittivity-DC electric fields in the supporting materials (Fig. S4).

Fig. R5. Hysteresis loops (a~c) and the ϵ_r -E curves (d~f) for the PZ thin films with thickness of 60 nm, 160 nm and 180 nm, respectively.

b) If there were two phases coexisting in the materials, it would be interesting to know if there are differences in the chemical compositions.

Response:

We thank for the reviewer's prudent consideration. We have characterized the elemental distribution of the thin films during this work and found no difference between two phases. As shown in Fig. R6, the EDS mapping images indicate identical composition for AFE-like and FiE-like phase. We add the EDS mapping in the supporting materials (Fig. S10).

Fig. R6. EDS mapping of the PZ thin films.

c) Standard theta-2theta scans are not sufficient to demonstrate good epitaxial growth.

Additional structural characterizations are required. In particular, reciprocal space mapping should be done to determine the actual lattice parameters of PbZrO_3 films.

Response:

We appreciate and accept the reviewer's suggestion. We select 180 nm PZ thin film to do both the atomic-scale HAADF and the reciprocal space mapping (RSM) for better determination of the epitaxial growth and the precise lattice parameters. On the one hand, the atomic-scale HAADF image presented in Fig. R3 exhibits the well-epitaxial PZ/STO interface. On the other hand, as shown in Fig. R7, the RSM plot is around the $(103)_c$ reflection of both STO substrate and PZO thin film. The two peaks of PZO mixed phases are labeled as AFE-like and FiE-like phase, respectively. The lattice parameters of AFE-like phase in PZO are: $a=b=4.154\text{\AA}$, $c=4.123\text{\AA}$, while for FiE-like phase they are: $a=b=4.163\text{\AA}$, $c=4.160\text{\AA}$. It can be seen that the parameters of AFE-like phase measured by RSM analysis are slightly smaller than that of FiE-like phase. We add the RSM in the supporting materials (Fig. S2) and the corresponding descriptions on page 7.

Fig. R7. Logarithmic reciprocal maps in the $H0L$ scattering plane around the $(103)_c$ reflections for the 180 nm PbZrO_3 thin films directly grown on SrTiO_3 substrate. (Intensities from low to high: blue-green-yellow-red-brown)

d) It appears that the FiE phase is more stable at zero strain in Figure 1A.

Response:

Although the energy of FiE phase at zero strain in Figure 1A is lower than AFE phase, the energy difference is too small. That is, the FiE phase may not be stabilized against the AFE phase by considering kinetic conditions. On the other hand, the theoretical results are calculated under 0 K, while the PZ thin films in the present study are

characterized at ambient temperature. The special strain state generated by the misfit dislocations and interfacial defects, i.e. APBs overall contribute to the stability of FiE phase as room temperature.

e) The streaking features observed for the diffraction patterns are not explained, and they are least visible in the 180-nm sample. Such features are not reflected in the modelled patterns.

Response:

The streaking features on diffraction patterns are originated from a high density of phase boundaries of FiE-like and AFE-like phase (please note that the streaks are oriented perpendicular to the phase boundaries). So, it is not reflected in the modelled patterns of single AFE-like or FiE-like phase. Weakening of streaking features refers to decreased density of phase boundaries due to coarsening of FiE-like crystallites. We added the corresponding explanation in the revised supporting information (the caption of Fig. S5).

Reference list

1. B. K. Mani, C.-M. Chang, S. Lisenkov, I. Ponomareva, Critical Thickness for Antiferroelectricity in PbZrO₃. *Phys. Rev. Lett.* **115**, 097601 (2015).
2. R. Gao, S. E. Reyes-Lillo, R. Xu, A. Dasgupta, Y. Dong, L. R. Dedon, J. Kim, S. Saremi, Z. Chen, C. R. Serrao, H. Zhou, J. B. Neaton, L. W. Martin, Ferroelectricity in Pb_{1+δ}ZrO₃ Thin Films. *Chem. Mater.* **29**, 6544–6551 (2017).
3. A. Roy Chaudhuri, M. Arredondo, A. Hähnel, A. Morelli, M. Becker, M. Alexe, I. Vrejoiu, Epitaxial strain stabilization of a ferroelectric phase in PbZrO₃ thin films. *Phys. Rev. B* **84**, 054112 (2011).
4. K. Boldyreva, D. Bao, G. Le Rhun, L. Pintilie, M. Alexe, D. Hesse, Microstructure and electrical properties of (120)O-oriented and of (001)O-oriented epitaxial antiferroelectric PbZrO₃ thin films on (100) SrTiO₃ substrates covered with different oxide bottom electrodes. *J. Appl. Phys.* **102**, 044111 (2007).
5. L. Pintilie, K. Boldyreva, M. Alexe, D. Hesse, Coexistence of ferroelectricity and antiferroelectricity in epitaxial PbZrO₃ films with different orientations. *J. Appl. Phys.* **103**, 024101 (2008).
6. Z. Fu, X. Chen, Z. Li, T. Hu, L. Zhang, P. Lu, S. Zhang, G. Wang, X. Dong, F. Xu, Unveiling the ferroelectric nature of PbZrO₃-based antiferroelectric materials. *Nat. Commun.* **11**, 3809 (2020).
7. X. Wei, K. Vaideeswaran, C. S. Sandu, C. Jia, N. Setter, Preferential Creation of Polar Translational Boundaries by Interface Engineering in Antiferroelectric PbZrO₃ Thin Films. *Adv. Mater. Interfaces* **2**, 1500349 (2015).
8. R.-J. Jiang, Y. Cao, W.-R. Geng, M.-X. Zhu, Y.-L. Tang, Y.-L. Zhu, Y. Wang, F. Gong, S.-Z. Liu, Y.-T. Chen, J. Liu, N. Liu, J.-H. Wang, X.-D. Lv, S.-J. Chen, X.-L. Ma,

Atomic Insight into the Successive Antiferroelectric–Ferroelectric Phase Transition in Antiferroelectric Oxides. *Nano Lett.* **23**, 1522–1529 (2023).

9. Y. Liu, R. Niu, A. Majchrowski, K. Roleder, K. Cordero-Edwards, J. M. Cairney, J. Arbiol, G. Catalan, Translational Boundaries as Incipient Ferrielectric Domains in Antiferroelectric PbZrO_3 . *Phys. Rev. Lett.* **130**, 216801 (2023).

10. Z. An, H. Yokota, K. Kurihara, N. Hasegawa, P. Marton, A. M. Glazer, Y. Uesu, W. Ren, Z. Ye, M. Paściak, N. Zhang, Tuning of Polar Domain Boundaries in Nonpolar Perovskite. *Adv. Mater.* **35**, 2207665 (2023).

11. T. Hu, Z. Fu, Z. Li, M. Liu, L. Zhang, Z. Yu, X. Chen, Y. Zheng, T. Li, Y. Wang, G. Wang, X. Dong, F. Xu, Decoding the Double/Multiple Hysteresis Loops in Antiferroelectric Materials. *ACS Appl. Mater. Interfaces* **13**, 60241–60249 (2021).

12. H. Aramberri, C. Cazorla, M. Stengel, and J. Íñiguez, On the possibility that PbZrO_3 not be antiferroelectric. *npj Comput. Mater.* **7**, 196 (2021).

13. Y. Yao, A. Naden, M. Tian, S. Lisenkov, Z. Beller, A. Kumar, J. Kacher, I. Ponomareva, N. Bassiri-Gharb, Ferrielectricity in the Archetypal Antiferroelectric, PbZrO_3 . *Adv. Mater.* **35**, 2206541 (2023).

14. R. G. Burkovsky, G. A. Lityagin, A. E. Ganzha, A. F. Vakulenko, R. Gao, A. Dasgupta, B. Xu, A. V. Filimonov, and L. W. Martin, Field-induced heterophase state in PbZrO_3 thin films. *Phys. Rev. B* **105**, 125409 (2022).

REVIEWERS' COMMENTS

Reviewer #1 (Remarks to the Author):

The article's authors have addressed my questions and made revisions to the manuscript. Overall, the quality of the article has improved. I believe the current version meets the high standards for publication in Nature Communications.

Reviewer #2 (Remarks to the Author):

The authors have addressed my concerns properly. I have no further question.

Reviewer #3 (Remarks to the Author):

In the revised version of the manuscript, the authors have addressed all my questions and concerns nicely. I would like to acknowledge the great efforts of the authors to conduct more experiments and calculations and revise the story and conclusions of the research. Therefore, I can recommend its publication in its current form.

Response Letter

Manuscript ID: NCOMMS-23-48001A

The authors sincerely thank the reviewers and the editor for consideration on the publication. And we appreciate it again for the positive comments and constructive suggestions from the reviewers. The point-by-point responses are listed below.

Responses to Reviewer #1

Comment:

The article's authors have addressed my questions and made revisions to the manuscript. Overall, the quality of the article has improved. I believe the current version meets the high standards for publication in Nature Communications.

Response:

We thank the reviewers for the highly recognition of our revision to the manuscript and the consideration of our manuscript for the publication.

Responses to Reviewer #2

Comment:

The authors have addressed my concerns properly.
I have no further question.

Response:

We thank the reviewers for the recognition and kind comments.

Responses to Reviewer #3

Comment:

In the revised version of the manuscript, the authors have addressed all my questions and concerns nicely. I would like to acknowledge the great efforts of the authors to conduct more experiments and calculations and revise the story and conclusions of the research. Therefore, I can recommend its publication in its current form.

Response:

We appreciate the reviewers' encouraging comments and kind recommendation of our manuscript, and we look forward to have more meaningful work in the future.